# GUIDING SAFE EXPLORATION WITH WEAKEST PRECONDITIONS

**Greg Anderson, Swarat Chaudhuri** [*]**Isil Dillig** [*]
Department of Computer Science
The University of Texas at Austin
Austin, TX, USA
{ganderso, swarat, isil}@cs.utexas.edu

## ABSTRACT

In reinforcement learning for safety-critical settings, it is often desirable for the agent to obey safety constraints at all points in time, including during training. We present a novel neurosymbolic approach called SPICE to solve this *safe exploration problem*. SPICE uses an online shielding layer based on symbolic *weakest preconditions* to achieve a more precise safety analysis than existing tools without unduly impacting the training process. We evaluate the approach on a suite of continuous control benchmarks and show that it can achieve comparable performance to existing safe learning techniques while incurring fewer safety violations. Additionally, we present theoretical results showing that SPICE converges to the optimal safe policy under reasonable assumptions.

## 1 INTRODUCTION

In many real-world applications of reinforcement learning (RL), it is crucial for the agent to behave safely during training. Over the years, a body of *safe exploration* techniques (Garcıa & Fernández, 2015) has emerged to address this challenge. Broadly, these methods aim to converge to high-performance policies while ensuring that every intermediate policy seen during learning satisfies a set of safety constraints. Recent work has developed neural versions of these methods (Achiam et al., 2017; Dalal et al., 2018; Bharadhwaj et al., 2021) that can handle continuous state spaces and complex policy classes.

Any method for safe exploration needs a mechanism for deciding if an action can be safely executed at a given state. Some existing approaches use prior knowledge about system dynamics (Berkenkamp et al., 2017; Anderson et al., 2020) to make such judgments. A more broadly applicable class of methods make these decisions using learned predictors represented as neural networks. For example, such a predictor can be a learned advantage function over the constraints (Achiam et al., 2017; Yang et al., 2020) or a critic network (Bharadhwaj et al., 2021; Dalal et al., 2018) that predicts the safety implications of an action.

However, neural predictors of safety can require numerous potentially-unsafe environment interactions for training and also suffer from approximation errors. Both traits are problematic in safety-critical, real-world settings. In this paper, we introduce a *neurosymbolic* approach to learning safety predictors that is designed to alleviate these difficulties.

Our approach, called SPICE [1], is similar to Bharadhwaj et al. (2021) in that we use a learned model to filter out unsafe actions. However, the novel idea in SPICE is to use the symbolic method of *weakest preconditions* (Dijkstra, 1976) to compute, from a single-time-step environment model, a predicate that decides if a given sequence of future actions is safe. Using this predicate, we symbolically compute a *safety shield* (Alshiekh et al., 2018) that intervenes whenever the current policy proposes an unsafe action. The environment model is repeatedly updated during the learning process using data safely collected using the shield. The computation of the weakest precondition and the shield is repeated, leading to a more refined shield, on each such update.

---

[*]equal advising
[1]SPICE is available at https://github.com/gavlegoat/spice.

The benefit of this approach is sample-efficiency: to construct a safety shield for the next $k$ time steps, SPICE only needs enough data to learn a single-step environment model. We show this benefit using an implementation of the method in which the environment model is given by a piecewise linear function and the shield is computed through quadratic programming (QP). On a suite of challenging continuous control benchmarks from prior work, SPICE has comparable performance as fully neural approaches to safe exploration and incurs far fewer safety violations on average.

In summary, this paper makes the following contributions:

- We present the first neurosymbolic framework for safe exploration with learned models of safety.
- We present a theoretical analysis of the safety and performance of our approach.
- We develop an efficient, QP-based instantiation of the approach and show that it offers greater safety than end-to-end neural approaches without a significant performance penalty.

## 2 PRELIMINARIES

**Safe Exploration.** We formalize safe exploration in terms of a constrained Markov decision process (CMDP) with a distinguished set of unsafe states. Specifically, a CMDP is a structure $\mathcal{M} = (\mathcal{S}, \mathcal{A}, r, P, p_0, c)$ where $\mathcal{S}$ is the set of states, $\mathcal{A}$ is the set of actions, $r : \mathcal{S} \times \mathcal{A} \to \mathbb{R}$ is a reward function, $P(\boldsymbol{x}' \mid \boldsymbol{x}, \boldsymbol{u})$, where $\boldsymbol{x}, \boldsymbol{x}' \in \mathcal{S}$ and $\boldsymbol{u} \in \mathcal{A}$, is a probabilistic transition function, $p_0$ is an initial distribution over states, and $c$ is a cost signal. Following prior work (Bharadhwaj et al., 2021), we consider the case where the cost signal is a boolean indicator of failure, and we further assume that the cost signal is defined by a set of *unsafe states* $\mathcal{S}_U$. That is, $c(\boldsymbol{x}) = 1$ if $\boldsymbol{x} \in \mathcal{S}_U$ and $c(\boldsymbol{x}) = 0$ otherwise. A *policy* is a stochastic function $\pi$ mapping states to distributions over actions. A policy, in interaction with the environment, generates *trajectories* (or *rollouts*) $\boldsymbol{x}_0, \boldsymbol{u}_0, \boldsymbol{x}_1, \boldsymbol{u}_1, \ldots, \boldsymbol{u}_{n-1}, \boldsymbol{x}_n$ where $\boldsymbol{x}_0 \sim p_0$, each $\boldsymbol{u}_i \sim \pi(\boldsymbol{x}_i)$, and each $\boldsymbol{x}_{i+1} \sim P(\boldsymbol{x}_i, \boldsymbol{u}_i)$. Consequently, each policy induces probability distributions $\mathcal{S}_\pi$ and $\mathcal{A}_\pi$ on the state and action. Given a *discount factor* $\gamma < 1$, the long-term *return* of a policy $\pi$ is $R(\pi) = \mathbb{E}_{\boldsymbol{x}_i, \boldsymbol{u}_i \sim \pi} \left[ \sum_i \gamma^i r(\boldsymbol{x}_i, \boldsymbol{u}_i) \right]$.

The goal of standard reinforcement learning is to find a policy $\pi^* = \arg\max_\pi R(\pi)$. Popular reinforcement learning algorithms accomplish this goal by developing a sequence of policies $\pi_0, \pi_1, \ldots, \pi_N$ such that $\pi_N \approx \pi^*$. We refer to this sequence of polices as a *learning process*. Given a bound $\delta$, the goal of *safe exploration* is to discover a learning process $\pi_0, \ldots, \pi_N$ such that

$$\pi_N = \arg\max_\pi R(\pi) \qquad \text{and} \qquad \forall 1 \le i \le N. \ P_{\boldsymbol{x} \sim \mathcal{S}_{\pi_i}}(\boldsymbol{x} \in \mathcal{S}_U) < \delta$$

That is, the final policy in the sequence should be optimal in terms of the long-term reward and every policy in the sequence (except for $\pi_0$) should have a bounded probability $\delta$ of unsafe behavior. Note that this definition does not place a safety constraint on $\pi_0$ because we assume that nothing is known about the environment a priori.

**Weakest Preconditions.** Our approach to the safe exploration problem is built on *weakest preconditions* (Dijkstra, 1976). At a high level, weakest preconditions allow us to "translate" constraints on a program's output to constraints on its input. As a very simple example, consider the function $x \mapsto x + 1$. The weakest precondition for this function with respect to the constraint $ret > 0$ (where $ret$ indicates the return value) would be $x > -1$. In this work, the "program" will be a model of the environment dynamics, with the inputs being state-action pairs and the outputs being states.

For the purposes of this paper, we present a simplified weakest precondition definition that is tailored towards our setting. Let $f : \mathcal{S} \times \mathcal{A} \to 2^{\mathcal{S}}$ be a nondeterministic transition function. As we will see in Section 4, $f$ represents a PAC-style bound on the environment dynamics. We define an alphabet $\Sigma$ which consists of a set of *symbolic* actions $\omega_0, \ldots, \omega_{H-1}$ and states $\chi_0, \ldots, \chi_H$. Each symbolic state and action can be thought of as a variable representing an *a priori unkonwn* state and action. Let $\phi$ be a first order formula over $\Sigma$. The symbolic states and actions represent a trajectory in the environment defined by $f$, so they are linked by the relation $\chi_{i+1} \in f(\chi_i, \omega_i)$ for $0 \le i < H$. Then, for a given $i$, the weakest precondition of $\phi$ is a formula $\psi$ over $\Sigma \setminus \{\chi_{i+1}\}$ such that (1) for all $e \in f(\chi_i, \omega_i)$, we have $\psi \implies \phi[\chi_{i+1} \mapsto e]$ and (2) for all $\psi'$ satisfying condition (1), $\psi' \implies \psi$. Here, the notation $\phi[\chi_{i+1} \mapsto e]$ represents the formula $\phi$ with all instances of $\chi_{i+1}$ replaced by the expression $e$. Intuitively, the first condition ensures that, after taking one environment step from $\chi_i$ under action $\omega_i$, the system will always satisfy $\phi$, no matter how the nondeterminism of $f$ is resolved. The second condition ensures that $\phi$ is as permissive as possible, which prevents us from ruling out states and actions that are safe in reality.

## 3    Symbolic Preconditions for Constrained Exploration

**Algorithm 1** The main learning algorithm

---

**procedure** SPICE
    Initialize an empty dataset $D$ and random policy $\pi$
    **for** epoch in $1 \ldots N$ **do**
        **if** epoch $= 1$ **then**
            $\pi_S \leftarrow \pi$
        **else**
            $\pi_S \leftarrow \lambda \boldsymbol{x}.\text{WPSHIELD}(M, \boldsymbol{x}, \pi(\boldsymbol{x}))^2$
        Unroll real trajectories $\{(s_i, a_i, s_i', r_i)\}$ under $\pi_S$
        $D = D \cup \{(s_i, a_i, s_i', r_i)\}$
        $M \leftarrow \text{LEARNENVMODEL}(D)$
        Optimize $\pi$ using the simulated environment $M$

---

Our approach, Symbolic Preconditions for Constrained Exploration (SPICE), uses a learned environment model to both improve sample efficiency and support safety analysis at training time. To do this, we build on top of model-based policy optimization (MBPO) (Janner et al., 2019). Similar to MBPO, the model in our approach is used to generate synthetic policy rollout data which can be fed into a model-free learning algorithm to train a policy. In contrast to MBPO, we reuse the environment model to ensure the safety of the system. This dual use of the environment allows for both efficient optimization and safe exploration.

The main training procedure is shown in Algorithm 1 and simultaneously learns an environment model $M$ and the policy $\pi$. The algorithm maintains a dataset $D$ of observed environment transitions, which is obtained by executing the current policy $\pi$ in the environment. SPICE then uses this dataset to learn an environment $M$, which is used to optimize the current policy $\pi$, as done in model-based RL. The key difference of our technique from standard model-based RL is the use of a *shielded policy* $\pi_S$ when unrolling trajectories to construct dataset $D$. This is necessary for safe exploration because executing $\pi$ in the real environment could result in safety violations. In contrast to prior work, the shielded policy $\pi_S$ in our approach is defined by an online weakest precondition computation which finds a *constraint* over the action space which *symbolically* represents all safe actions. This procedure is described in detail in Section 4.

## 4    Shielding with Polyhedral Weakest Preconditions

### 4.1    Overview of Shielding Approach

**Algorithm 2** Shielding a proposed action

---

**procedure** WPSHIELD($M, \boldsymbol{x}_0, \boldsymbol{u}_0^*$)
    $f \leftarrow \text{APPROXIMATE}(M, \boldsymbol{x}_0, \boldsymbol{u}_0^*)$
    $\phi_H \leftarrow \bigwedge_{i=1}^{H} \chi_i \in \mathcal{S} \setminus \mathcal{S}_U$
    **for** $t$ from $H - 1$ down to $0$ **do**
        $\phi_t \leftarrow \text{WP}(\phi_{t+1}, f)$
    $\phi \leftarrow \phi_0[\chi_0 \mapsto \boldsymbol{x}_0]$
    $(\boldsymbol{u}_0, \ldots, \boldsymbol{u}_{H-1}) = \underset{\boldsymbol{u}_0', \ldots, \boldsymbol{u}_{H-1}' \vDash \phi}{\arg\min} \|\boldsymbol{u}_0' - \boldsymbol{u}_0^*\|^2$
    **return** $\boldsymbol{u}_0$

---

Our high-level online intervention approach is presented in Algorithm 2. Given an environment model $M$, the current state $\boldsymbol{x}_0$ and a proposed action $\boldsymbol{u}_0^*$, the WPSHIELD procedure chooses a modified action $\boldsymbol{u}_0$ which is as similar as possible to $\boldsymbol{u}_0^*$ while ensuring safety. We consider an action to be *safe* if, after executing that action in the environment, there exists a sequence of follow-up actions $\boldsymbol{u}_1, \ldots, \boldsymbol{u}_{H-1}$ which keeps the system away from the unsafe states over a finite time horizon $H$. In more detail, our intervention technique works in three steps:

**Approximating the environment.** Because computing the weakest precondition of a constraint with respect to a complex environment model (e.g., deep neural network) is intractable, Algorithm 2 calls the APPROXIMATE procedure to obtain a simpler first-order local Taylor approximation to the environment model centered at $(\boldsymbol{x}_0, \boldsymbol{u}_0^*)$. That is, given the environment model $M$, it computes matrices $\boldsymbol{A}$ and $\boldsymbol{B}$, a vector $\boldsymbol{c}$, and an error $\varepsilon$ such that $f(\boldsymbol{x}, \boldsymbol{u}) = \boldsymbol{A}\boldsymbol{x} + \boldsymbol{B}\boldsymbol{u} + \boldsymbol{c} + \Delta$ where $\Delta$ is an unknown vector with elements in $[-\varepsilon, \varepsilon]$. The error term is computed based on a normal Taylor series analysis such that with high probability, $M(\boldsymbol{x}, \boldsymbol{u}) \in f(\boldsymbol{x}, \boldsymbol{u})$ in a region close to $\boldsymbol{x}_0$ and $\boldsymbol{u}_0^*$.

**Computation of safety constraint.** Given a linear approximation $f$ of the environment, Algorithm 2 iterates backwards in time, starting with the safety constraint $\phi_H$ at the end of the time horizon $H$.

---

[2]Note that $\lambda$ is an anonymous function operator rather than a regularization constant.

In particular, the initial constraint $\phi_H$ asserts that all (symbolic) states $\chi_1, \ldots, \chi_H$ reached within the time horizon are inside the safe region. Then, the loop inside Algorithm 2 uses the WP procedure (described in the next two subsections) to eliminate one symbolic state at a time from the formula $\phi_i$. After the loop terminates, all of the state variables except for $\chi_0$ have been eliminated from $\phi_0$, so $\phi_0$ is a formula over $\chi_0, \omega_0, \ldots, \omega_{H-1}$. The next line of Algorithm 2 simply replaces the symbolic variable $\chi_0$ with the current state $\boldsymbol{x}_0$ in order to find a constraint over only the actions.

**Projection onto safe space.** The final step of the shielding procedure is to find a sequence $\boldsymbol{u}_0, \ldots, \boldsymbol{u}_{H-1}$ of actions such that (1) $\phi$ is satisfied and (2) the distance $\|\boldsymbol{u}_0 - \boldsymbol{u}_0^*\|$ is minimized. Here, the first condition enforces the safety of the shielded policy, while the second condition ensures that the shielded policy is as similar as possible to the original one. The notation $\boldsymbol{u}_0, \ldots, \boldsymbol{u}_{H-1} \vDash \phi$ indicates that $\phi$ is true when the concrete values $\boldsymbol{u}_0, \ldots, \boldsymbol{u}_{H-1}$ are substituted for the symbolic values $\omega_0, \ldots, \omega_{H-1}$ in $\phi$. Thus, the $\arg\min$ in Algorithm 2 is effectively a projection on the set of action sequences satisfying $\phi$. We discuss this optimization problem in Section 4.4.

## 4.2 WEAKEST PRECONDITIONS FOR POLYHEDRA

In this section, we describe the WP procedure used in Algorithm 2 for computing the weakest precondition of a safety constraint $\phi$ with respect to a linear environment model $f$. To simplify presentation, we assume that the safe space is given as a convex polyhedron — i.e., all safe states satisfy the linear constraint $\boldsymbol{P}\boldsymbol{x} + \boldsymbol{q} \le \boldsymbol{0}$. We will show how to relax this restriction in Section 4.3.

Recall that our environment approximation $f$ is a linear function with bounded error, so we have constraints over the symbolic states and actions: $\chi_{i+1} = \boldsymbol{A}\chi_i + \boldsymbol{B}\omega_i + \boldsymbol{c} + \Delta$ where $\Delta$ is an *unknown* vector with elements in $[-\varepsilon, \varepsilon]$. In order to compute the weakest precondition of a linear constraint $\phi$ with respect to $f$, we simply replace each instance of $\chi_{i+1}$ in $\phi$ with $\boldsymbol{A}\chi_i + \boldsymbol{B}\omega_i + \boldsymbol{c} + \Delta^*$ where $\Delta^*$ is the *most pessimistic* possibility for $\Delta$. Because the safety constraints are linear and the expression for $\chi_{i+1}$ is also linear, this substitution results in a new linear formula which is a conjunction of constraints of the form $\boldsymbol{w}^T\nu + \boldsymbol{v}^T\Delta^* \le y$. For each element $\Delta_i$ of $\Delta$, if the coefficient of $\Delta_i^*$ is positive in $\boldsymbol{v}$, then we choose $\Delta_i^* = \varepsilon$. Otherwise, we choose $\Delta_i^* = -\varepsilon$. This substitution yields the maximum value of $\boldsymbol{v}^T\Delta^*$ and is therefore the most pessimistic possibility for $\Delta_i^*$.

**Example.** We illustrate the weakest precondition computation through simple example: Consider a car driving down a (one-dimensional) road whose goal is to reach the other end of the road as quickly as possible while obeying a speed limit. The state of the car is a position $x$ and velocity $v$. The action space consists of an acceleration $a$. Assume there is bounded noise in the velocity updates so the dynamics are $x' = x + 0.1v$ and $v' = v + 0.1a + \varepsilon$ where $-0.01 \le \varepsilon \le 0.01$ and the safety constraint is $v \le 1$. Suppose the current velocity is $v_0 = 0.9$ and the safety horizon is two. Then, starting with the safety constraint $v_1 \le 1 \wedge v_2 \le 1$ and stepping back through the environment dynamics, we get the precondition $v_1 \le 1 \wedge v_1 + 0.1a_1 + \varepsilon_1 \le 1$. Stepping back one more time, we find the condition

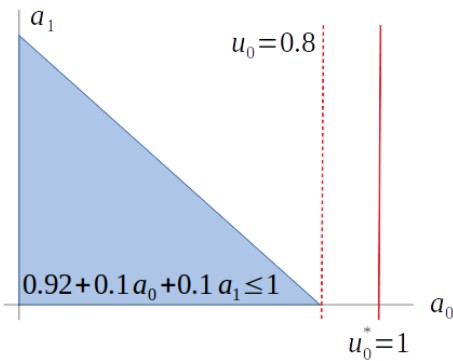

Figure 1: Weakest precondition example.

$v_0 + 0.1a_0 + \varepsilon_2 \le 1 \wedge v_0 + 0.1a_0 + 0.1a_1 + \varepsilon_1 + \varepsilon_2 \le 1$. Picking the most pessimistic values for $\varepsilon_1$ and $\varepsilon_2$ to reach $v_0 + 0.1a_0 + 0.01 \le 1 \wedge v_0 + 0.1a_0 + 0.1a_1 + 0.02 \le 1$. Since $v_0$ is specified, we can replace $v_0$ with 0.9 to simplify this to a constraint over the two actions $a_0$ and $a_1$, namely $0.91 + 0.1a_0 \le 1 \wedge 0.92 + 0.1a_0 + 0.1a_1 \le 1$. Figure 1 shows this region as the shaded triangle on the left. Any pair of actions $(a_0, a_1)$ which lies inside the shaded triangle is guaranteed to satisfy the safety condition for any possible values of $\varepsilon_1$ and $\varepsilon_2$.

## 4.3 EXTENSION TO MORE COMPLEX SAFETY CONSTRAINTS

In this section, we extend our weakest precondition computation technique to the setting where the safe region consists of *unions* of convex polyhedra. That is, the state space is represented as a set of matrices $\boldsymbol{P}_i$ and a set of vectors $\boldsymbol{q}_i$ such that $\mathcal{S} \setminus \mathcal{S}_U = \bigcup_{i=1}^N \{\boldsymbol{x} \in \mathcal{S} \mid \boldsymbol{P}_i\boldsymbol{x} + \boldsymbol{q}_i \le \boldsymbol{0}\}$.

Note that, while individual polyhedra are limited in their expressive power, unions of polyhedra can approximate reasonable spaces with arbitrary precision. This is because a single polyhedron can approximate a convex set arbitrarily precisely (Bronshteyn & Ivanov, 1975), so unions of polyhedra can approximate unions of convex sets.

In this case, the formula $\phi_H$ in Algorithm 2 has the form $\phi_H = \bigwedge_{j=1}^{H} \bigvee_{i=1}^{N} \boldsymbol{P}_i \chi_j + \boldsymbol{q}_i \leq \boldsymbol{0}$. However, the weakest precondition of a formula of this kind can be difficult to compute. Because the system may transition between two different polyhedra at each time step, there is a combinatorial explosion in the size of the constraint formula, and a corresponding exponential slowdown in the weakest precondition computation. Therefore, we replace $\phi_H$ with an approximation $\phi'_H = \bigvee_{i=1}^{N} \bigwedge_{j=1}^{H} \boldsymbol{P}_i \chi_j + \boldsymbol{q}_i \leq \boldsymbol{0}$ (that is, we swap the conjunction and the disjunction). Note that $\phi_H$ and $\phi'_H$ are *not* equivalent, but $\phi'_H$ is a stronger formula (i.e., $\phi'_H \implies \phi_H$). Thus, any states satisfying $\phi'_H$ are also guaranteed to satisfy $\phi_H$, meaning that they will be safe. More intuitively, this modification asserts that, not only does the state stay within the safe region at each time step, but it stays within *the same polyhedron* at each step within the time horizon.

With this modified formula, we can pull the disjunction outside the weakest precondition, i.e.,

$$\mathrm{WP}\left(\bigvee_{i=1}^{N} \bigwedge_{j=1}^{H} \boldsymbol{P}_i \chi_j + \boldsymbol{q}_i \leq \boldsymbol{0}, f\right) = \bigvee_{i=1}^{N} \mathrm{WP}\left(\bigwedge_{j=1}^{H} \boldsymbol{P}_i \chi_j + \boldsymbol{q}_i \leq \boldsymbol{0}, f\right).$$

The conjunctive weakest precondition on the right is of the form described in Section 4.2, so this computation can be done efficiently. Moreover, the number of disjuncts does not grow as we iterate through the loop in Algorithm 2. This prevents the weakest precondition formula from growing out of control, allowing for the overall weakest precondition on $\phi'_H$ to be computed quickly.

Intuitively, the approximation we make to the formula $\phi_H$ does rule out some potentially safe action sequences. This is because it requires the system to stay within *a single* polyhedron over the entire horizon. However, this imprecision can be ameliorated in cases where the different polyhedra comprising the state space overlap one another (and that overlap has non-zero volume). In that case, the overlap between the polyhedra serves as a "transition point," allowing the system to maintain safety within one polyhedron until it enters the overlap, and then switch to the other polyhedron in order to continue its trajectory. A formal development of this property, along with an argument that it is satisfied in many practical cases, is laid out in Appendix B.

**Example.** Consider an environment which represents a robot moving in 2D space. The state space is four-dimensional, consisting of two position elements $x$ and $y$ and two velocity elements $v^x$ and $v^y$. The action space consists of two acceleration terms $a^x$ and $a^y$, giving rise to the dynamics

$$x = x + 0.1v^x \quad y = y + 0.1v^y$$
$$v^x = v^x + 0.1a^x \quad v^y = v^y + 0.1a^y$$

In this environment, the safe space is $x \geq 2 \vee y \leq 1$, so that the upper-left part of the state space is considered unsafe. Choosing a safety horizon of $H = 2$, we start with the initial constraint $(x_1 \geq 2 \vee y_1 \leq 1) \wedge (x_1 \geq 2 \wedge y_2 \leq 1)$. We transform this formula to the stronger formula $(x_1 \geq 2 \wedge x_2 \geq 2) \vee (y_1 \leq 1 \wedge y_2 \leq 1)$. By stepping backwards through the weakest precondition twice, we obtain the following formula over only the current state and future actions:

$$(x_0 + 0.1v_0^x \geq 2 \wedge x_0 + 0.2v_0^x + 0.01a_0^x \geq 2) \vee (y_0 + 0.1v_0^y \leq 1 \wedge y_0 + 0.2v_0^y + 0.01a_0^y \leq 1).$$

## 4.4 Projection Onto the Weakest Precondition

After applying the ideas from Section 4.3, each piece of the safe space yields a set of linear constraints over the action sequence $\boldsymbol{u}_0, \ldots, \boldsymbol{u}_{H-1}$. That is, $\phi$ from Algorithm 2 has the form

$$\phi = \bigvee_{i=1}^{N} \sum_{j=0}^{H-1} \boldsymbol{G}_{i,j} \boldsymbol{u}_j + \boldsymbol{h}_i \leq 0.$$

Now, we need to find the action sequence satisfying $\phi$ for which the first action most closely matches the proposed action $\boldsymbol{u}_0^*$. In order to do this, we can minimize the objective function $\|\boldsymbol{u}_0 - \boldsymbol{u}_0^*\|^2$. This function is quadratic, so we can represent this minimization problem as $N$ quadratic programming problems. That is, for each polyhedron $\boldsymbol{P}_i, \boldsymbol{q}_i$ in the safe region, we solve:

$$\begin{aligned} \text{minimize} \quad & \|\boldsymbol{u}_0^* - \boldsymbol{u}_0\|^2 \\ \text{subject to} \quad & \sum_{j=0}^{H-1} \boldsymbol{G}_{i,j} \boldsymbol{u}_j + \boldsymbol{h}_i \leq \boldsymbol{0} \end{aligned}$$

Such problems can be solved efficiently using existing tools. By applying the same technique independently to each piece of the safe state space, we reduce the projection problem to a relatively small number of calls to a quadratic programming solver. This reduction allows the shielding procedure to be applied fast enough to generate the amount of data needed for gradient-based learning.

**Example:** Consider again Figure 1. Suppose the proposed action is $\boldsymbol{u}_0^* = 1$, represented by the solid line in Figure 1. Since the proposed action is outside of the safe region, the projection operation will find the point inside the safe region that minimizes the distance *along the $a_0$ axis only*. This leads to the dashed line in Figure 1, which is the action $\boldsymbol{u}_0$ that is as close as possible to $\boldsymbol{u}_0^*$ while still intersecting the safe region represented by the shaded triangle. Therefore, in this case, WPSHIELD would return 0.8 as the safe action.

## 5 THEORETICAL RESULTS

We will now develop theoretical results on the safety and performance of agents trained with SPICE. For brevity, proofs have been deferred to Appendix A.

For the safety theorem, we will assume the model is approximately accurate with high probability and that the APPROXIMATE procedure gives a sound local approximation to the model. Formally, $\Pr_{\boldsymbol{x}' \sim P(\cdot | \boldsymbol{x}, \boldsymbol{u})}[\|M(\boldsymbol{x}, \boldsymbol{u}) - \boldsymbol{x}'\| > \varepsilon] < \delta_M$, and if $f = \text{APPROXIMATE}(M, \boldsymbol{x}_0, \boldsymbol{u}_0^*)$ then for all actions $\boldsymbol{u}$ and all states $\boldsymbol{x}$ reachable within $H$ time steps, $M(\boldsymbol{x}, \boldsymbol{u}) \in f(\boldsymbol{x}, \boldsymbol{u})$.

**Theorem 1.** *Let $\boldsymbol{x}_0$ be a safe state and let $\pi$ be any policy. For $0 \leq i < H$, let $\boldsymbol{u}_i = \text{WPSHIELD}(M, \boldsymbol{x}_i, \pi(\boldsymbol{x}_i))$ and let $\boldsymbol{x}_{i+1}$ be the result of taking action $\boldsymbol{u}_i$ at state $\boldsymbol{x}_i$. Then with probability at least $(1 - \delta_M)^i$, $\boldsymbol{x}_i$ is safe.*

This theorem shows why SPICE is better able to maintain safety compared to prior work. Intuitively, constraint violations can only occur in SPICE when the environment model is incorrect. In contrast, statistical approaches to safe exploration are subject to safety violations caused by *either* modeling error *or* actions which are not safe even with respect to the environment model. Note that for a safety level $\delta$ and horizon $H$, a modeling error can be computed as $\delta_M < 1 - (1 - \delta)/\exp(H - 1)$.

The performance analysis is based on treating Algorithm 1 as a functional mirror descent in the policy space, similar to Verma et al. (2019) and Anderson et al. (2020). We assume a class of neural policies $\mathcal{F}$, a class of safe policies $\mathcal{G}$, and a joint class $\mathcal{H}$ of neurosymbolic policies. We proceed by considering the shielded policy $\lambda\boldsymbol{x}.\text{WPSHIELD}(M, \boldsymbol{x}, \pi_N(\boldsymbol{x}))$ to be a *projection* of the neural policy $\pi_N$ into $\mathcal{G}$ for a Bregman divergence $D_F$ defined by a function $F$. We define a safety indicator $Z$ which is one whenever $\text{WPSHIELD}(M, \boldsymbol{x}, \pi^{(i)}(\boldsymbol{x})) = \pi^{(i)}(\boldsymbol{x})$ and zero otherwise, and we let $\zeta = \mathbb{E}[1 - Z]$. Under reasonable assumptions (see Appendix A for a full discussion), we prove a regret bound for Algorithm 1.

**Theorem 2.** *Let $\pi_S^{(i)}$ for $1 \leq i \leq T$ be a sequence of safe policies learned by SPICE (i.e., $\pi_S^{(i)} = \lambda\boldsymbol{x}.\text{WPSHIELD}(M, \boldsymbol{x}, \pi(\boldsymbol{x})))$ and let $\pi_S^*$ be the optimal safe policy. Additionally we assume the reward function $R$ is Lipschitz in the policy space and let $L_R$ be the Lipschitz constant of $R$, $\beta$ and $\sigma^2$ be the bias and variance introduced by sampling in the gradient computation, $\epsilon$ be an upper bound on the bias incurred by using projection onto the weakest precondition to approximate imitation learning, $\epsilon_m$ be an upper bound the KL divergence between the model and the true environment dynamics at all time steps, and $\epsilon_\pi$ be an upper bound on the TV divergence between the policy used to gather data and the policy being trained at all time steps. Then setting the learning rate $\eta = \sqrt{\frac{1}{\sigma^2}\left(\frac{1}{T} + \epsilon\right)}$, we have the expected regret bound:*

$$R\left(\pi_S^*\right) - \mathbb{E}\left[\frac{1}{T}\sum_{i=1}^{T} R\left(\pi_S^{(i)}\right)\right] = O\left(\sigma\sqrt{\frac{1}{T} + \epsilon} + \beta + L_R\zeta + \epsilon_m + \epsilon_\pi\right)$$

This theorem provides a few intuitive results, based on the additive terms in the regret bound. First, $\zeta$ is the frequency with which we intervene in network actions and as $\zeta$ decreases, the regret bound becomes tighter. This fits our intuition that, as the shield intervenes less and less, we approach standard reinforcement learning. The two terms $\epsilon_m$ and $\epsilon_\pi$ are related to how accurately the model captures the true environment dynamics. As the model becomes more accurate, the policy converges to better returns. The other terms are related to standard issues in reinforcement learning, namely the error incurred by using sampling to approximate the gradient.

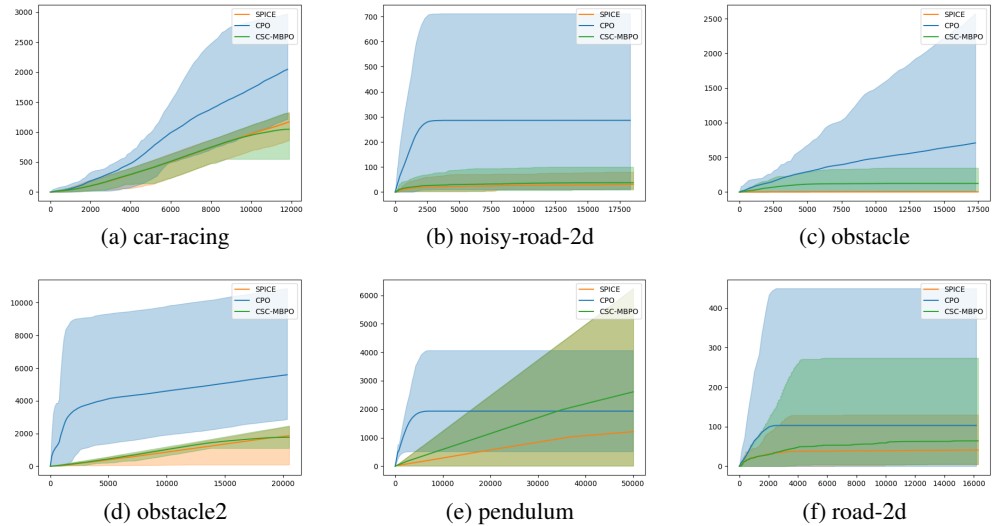

Figure 2: Cumulative safety violations over time.

# 6 EXPERIMENTAL EVALUATION

We now turn to a practical evaluation of SPICE. Our implementation of SPICE uses PyEarth (Rudy, 2013) for model learning and CVXOPT (Anderson et al., 2022) for quadratic programming. Our learning algorithm is based on MBPO (Janner et al., 2019) using Soft Actor-Critic (Haarnoja et al., 2018a) as the underlying model-free learning algorithm. Our code is adapted from Tandon (2018). We test SPICE using the benchmarks considered in Anderson et al. (2020). Further details of the benchmarks and hyperparameters are given in Appendix C.

We compare against two baseline approaches: Constrained Policy Optimization (CPO) (Achiam et al., 2017), a model-free safe learning algorithm, and a version of our approach which adopts the conservative safety critic shielding framework from Bharadhwaj et al. (2021) (CSC-MBPO). Details of the CSC-MBPO approach are given in Appendix C. We additionally tested MPC-RCE (Liu et al., 2020), another model-based safe-learning algorithm, but we find that it is too inefficient to be run on our benchmarks. Specifically MPC-RCE was only able to finish on average 162 episodes within a 2-day time period. Therefore, we do not include MPC-RCE in the results presented in this section.

| Benchmark | CPO | CSC-MBPO | SPICE |
|---|---|---|---|
| acc | 684 | 137 | 286 |
| car-racing | 2047 | 1047 | 1169 |
| mountain-car | 2374 | 2389 | 6 |
| noisy-road | 0 | 0 | 0 |
| noisy-road-2d | 286 | 37 | 31 |
| obstacle | 708 | 124 | 2 |
| obstacle2 | 5592 | 1773 | 1861 |
| pendulum | 1933 | 2610 | 1211 |
| road | 0 | 0 | 0 |
| road-2d | 103 | 64 | 41 |
| Average | 9.48 | 3.77 | 1 |

Table 1: Safety violations during training.

**Safety.** First, we evaluate how well our approach ensures system safety during training. In Table 1, we present the number of safety violations encountered during training for our baselines. The last row of the table shows the average increase in the number of safety violations compared to SPICE (computed as the geometric mean of the ratio of safety violations for each benchmark). This table shows that SPICE is safer than CPO in every benchmark and achieves, on average, a 89% reduction in safety violations. CSC-MBPO is substantially safer than CPO, but still not as safe as SPICE. We achieve a 73% reduction in safety violations on average compared to CSC-MBPO. To give a more detailed breakdown, Figure 2 shows how the safety violations accumulate over time for several of our benchmarks. The solid line represents the mean over all trials while the shaded envelope shows the minimum and maximum values. As can be seen from these figures, CPO starts to accumulate violations more quickly and continues to violate the safety property more over time than SPICE. Figures for the remaining benchmarks can be found in Appendix C.

Note that there are a few benchmarks (acc, car-racing, and obstacle2) where SPICE incurs more violations than CSC-MBPO. There are two potential reasons for this increase. First, SPICE relies

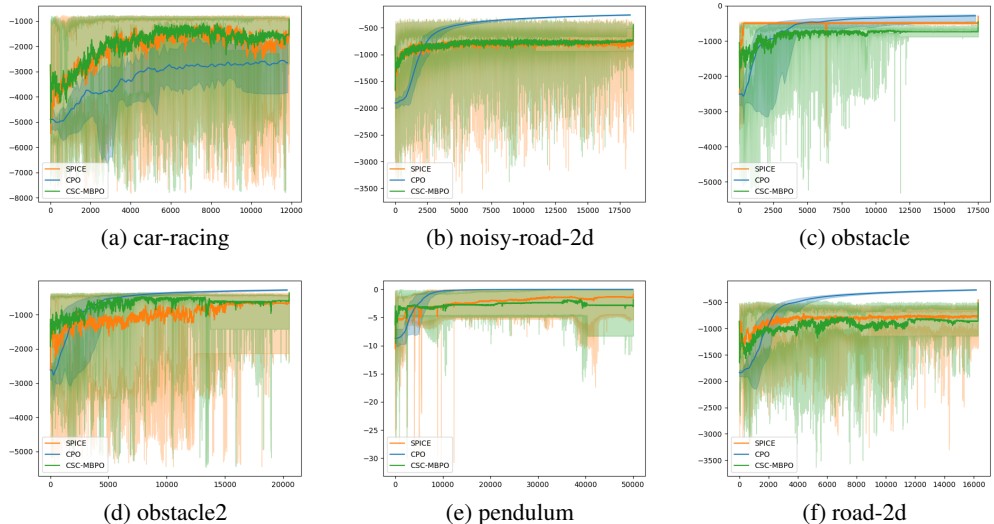

(a) car-racing  (b) noisy-road-2d  (c) obstacle

(d) obstacle2  (e) pendulum  (f) road-2d

Figure 3: Training curves for SPICE and CPO.

on choosing a model class through which to compute weakest preconditions (i.e., we need to fix an APPROXIMATE function in Algorithm 2). For these experiments, we use a linear approximation, but this can introduce a lot of approximation error. A more complex model class allowing a more precise weakest precondition computation may help to reduce safety violations. Second, SPICE uses a bounded-time analysis to determine whether a safety violation can occur within the next few time steps. By contrast, CSC-MBPO uses a neural model to predict the long-term safety of an action. As a result, actions which result in safety violations far into the future may be easier to intercept using the CSC-MBPO approach. Given that SPICE achieves much lower safety violations on average, we think these trade-offs are desirable in many situations.

**Performance.** We also test the performance of the learned policies on each benchmark in order to understand what impact our safety techniques have on model learning. Figure 3 show the average return over time for SPICE and the baselines. These curves show that in most cases SPICE achieves a performance close to that of CPO, and about the same as CSC-MBPO. We believe that the relatively modest performance penalty incurred by SPICE is an acceptable trade-off in some safety-critical systems given the massive improvement in safety. Further results are presented in Appendix C.

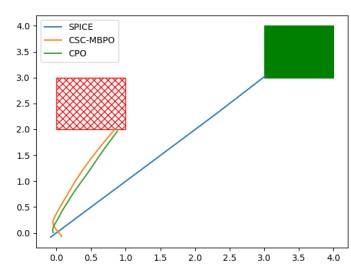

Figure 4: Trajectories early in training.

**Qualitative Evaluation.** Figure 4 shows the trajectories of policies learned by SPICE, CPO, and CSC-MBPO partway through training (after 300 episodes). In this figure, the agent controls a robot moving on a 2D plane which must reach the green shaded area while avoiding the red cross-hatched area. For each algorithm, we sampled 100 trajectories and plotted the worst one. Note that, at this point during training, both CPO and CSC-MBPO behave unsafely, while the even worst trajectory sampled under SPICE was still safe. See Appendix C for a more complete discussion of how these trajectories evolve over time during training.

## 7  RELATED WORK

Existing work in safe reinforcement learning can be categorized by the kinds of guarantees it provides: statistical approaches bound the probability that a violation will occur, while worst-case analyses prove that a policy can never reach an unsafe state. SPICE is, strictly speaking, a statistical approach—without a predefined environment model, we cannot guarantee that the agent will

never behave unsafely. However, as we show experimentally, our approach is substantially safer in practice than existing safe learning approaches based on learned cost models.

**Statistical Approaches.** Many approaches to the safe reinforcement learning problem provide statistical bounds on the system safety (Achiam et al., 2017; Liu et al., 2020; Yang et al., 2020; Ma et al., 2021; Zhang et al., 2020; Satija et al., 2020). These approaches maintain an environment model and then use a variety of statistical techniques to generate a policy which is likely to be safe with respect to the environment model. This leads to two potential sources of unsafe behavior: the policy may be unsafe with respect to the model, or the model may be an inaccurate representation of the environment. Compared to these approaches, we eliminate the first source of error by always generating policies that are *guaranteed* to be safe with respect to an environment model. We show in our experiments that this drastically reduces the number of safety violations encountered in practice. Some techniques use a learned model together with a linearized cost model to provide bounds, similar to our approach (Dalal et al., 2018; Li et al., 2021). However, in these works, the system only looks ahead one time step and relies on the assumption that the cost signal cannot have too much inertia. Our work alleviates this problem by providing a way to look ahead several time steps to achieve a more precise safety analysis.

A subset of the statistical approaches are tools that maintain neural models of the cost function in order to intervene in unsafe behavior (Bharadhwaj et al., 2021; Yang et al., 2021; Yu et al., 2022). These approaches maintain a critic network which represents the long-term cost of taking a particular action in a particular state. However, because of the amount of data needed to train neural networks accurately, these approaches suffer from a need to collect data in several unsafe trajectories in order to build the cost model. Our symbolic approach is more data-efficient, allowing the system to avoid safety violations more often in practice. This is borne out by the experiments in Section 6.

**Worst-Case Approaches.** Several existing techniques for safe reinforcement learning provide formally verified guarantees with respect to a worst-case environment model, either during training (Anderson et al., 2020) or at convergence (Alshiekh et al., 2018; Bacci et al., 2021; Bastani et al., 2018; Fulton & Platzer, 2019; Gillula & Tomlin, 2012; Zhu et al., 2019). An alternative class of approaches uses either a *nominal* environment model (Koller et al., 2018; Fisac et al., 2019) or a user-provided safe policy as a starting point for safe learning (Chow et al., 2018; Cheng et al., 2019). In both cases, these techniques require a predefined model of the dynamics of the environment. In contrast, our technique does not require the user to specify any model of the environment, so it can be applied to a much broader set of problems.

# 8 CONCLUSION

SPICE is a new approach to safe exploration that combines the advantages of gradient-based learning with symbolic reasoning about safety. In contrast to prior work on formally verified exploration (Anderson et al., 2020), SPICE can be used without a precise, handwritten specification of the environment behavior. The linchpin of our approach is a new policy intervention which can efficiently intercept unsafe actions and replace them with actions which are as similar as possible, but provably safe. This intervention method is fast enough to support the data demands of gradient-based reinforcement learning and precise enough to allow the agent to explore effectively.

There are a few limitations to SPICE. Most importantly, because the interventions are based on a linearized environment model, they are only accurate in a relatively small region near the current system state. This in turn limits the time horizons which can be considered in the safety analysis, and therefore the strength of the safety properties. Our experiments show that SPICE is still able to achieve good empirical safety in this setting, but a more advanced policy intervention that can handle more complex environment models could further improve these results. Additionally, SPICE can make no safety guarantees about the initial policy used to construct the first environment model, since there is no model to verify against at the time that that policy is executed. This issue could be alleviated by assuming a conservative initial model which is refined over time, and there is an interesting opportunity for future work combining partial domain expertise with learned dynamics.

FUNDING ACKNOWLEDGEMENTS

This work was supported in part by the United States Air Force and DARPA under Contract No. FA8750-20-C-0002, by ONR under Award No. N00014-20-1-2115, and by NSF under grants CCF-1901376 and CCF-1918889. Compute resources for the experiments were provided by the Texas Advanced Computing Center.

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

## A  PROOFS OF THEOREMS

In this section we present proofs for the theorems in Section 5. First, we will look at the safety results. We need some assumptions for this theorem:

**Assumption 1.** *The function* APPROXIMATE *returns a sound, nondeterministic approximation of $M$ in a region reachable from state $x_0$ over a time horizon $H$. That is, let $\mathcal{S}_R$ be the set of all states $x$ for which there exists a sequence of actions under which the system can transition from $x_0$ to $x$ within $H$ time steps. Then if $f = $ APPROXIMATE$(M, x_0, u_0^*)$ then for all $x \in \mathcal{S}_R$ and $u \in \mathcal{A}$, $M(x, u) \in f(x, u)$.*

**Assumption 2.** *The model learning procedure returns a model which is close to the actual environment with high probability. That is, if $M$ is a learned environment model then for all $x, u$,*

$$\Pr_{x' \sim P(\cdot | x, u)} \left[ \|M(x, u) - x'\| > \varepsilon \right] < \delta$$

**Definition 1.** *A state $x_0$ is said to have* realizable safety *over a time horizon $H$ if there exists a sequence of actions $u_0, \ldots, u_{H-1}$ such that, when $x_0, \ldots, x_H$ is the trajectory unrolled starting from $x_0$ in the true environment, the formula $\phi$ inside* WPSHIELD$(M, x_i, \pi(x_i))$ *is satisfiable for all $i$.*

**Lemma 1.** *(*WPSHIELD *is safe under bounded error.) Let $H$ be a time horizon, $x_0$ be a state with realizable safety over $H$, $M$ be an environment model, and $\pi$ be a policy. Choose $\varepsilon$ such that for all states $x$ and actions $u$, $\|M(x, u) - x'\| \leq \varepsilon$ where $x'$ is sampled from the true environment transition at $x$ and $u$. For $0 \leq i < H$ let $u_i = $ WPSHIELD$(M, x_i, \pi(x_i))$ and let $x_{i+1}$ be sampled from the true environment at $x_i$ and $u_i$. Then for $0 \leq i \leq H$, $x_i \notin \mathcal{S}_U$.*

*Proof.* Combining Assumption 1 with condition 1 in the definition of weakest preconditions, we conclude that for all $e \in M(\boldsymbol{x}_i, \boldsymbol{u}_i)$, $\text{WP}(\phi_{i+1}, f) \implies \phi_{i+1}[\boldsymbol{x}_{i+1} \mapsto e]$. Stepping backward through the loop in WPSHIELD, we find that for all $e_i \in M(\boldsymbol{x}_{i-1}, \boldsymbol{u}_{i-1})$ for $1 \leq i \leq H$, $\phi_0 \implies \phi_H[\boldsymbol{x}_1 \mapsto e_1, \ldots, \boldsymbol{x}_H \mapsto e_H]$. Because $\phi_H$ asserts the safety of the system, we have that $\phi_0$ also implies that the system is safe. Then because the actions returned by WPSHIELD are constrained to satisfy $\phi_0$, we also have that $\boldsymbol{x}_i$ for $0 \leq i \leq H$ are safe. $\qquad\square$

**Theorem 1.** *(WPSHIELD is probabilistically safe.) For a given state $\boldsymbol{x}_0$ with realizable safety over a time horizon $H$, a neural policy $\pi$, and an environment model $M$, let $\varepsilon$ and $\delta$ be the error and probability bound of the model as defined in Assumption 2. For $0 \leq i < H$, let $\boldsymbol{u}_i = \text{WPSHIELD}(M, \boldsymbol{x}_i, \pi(\boldsymbol{x}_i))$ and let $\boldsymbol{x}_{i+1}$ be the result of taking action $\boldsymbol{u}_i$ at state $\boldsymbol{x}_i$ then with probability at least $(1 - \delta)^i$, $\boldsymbol{x}_i$ is safe.*

*Proof.* By Lemma 1, if $\|M(\boldsymbol{x}, \boldsymbol{u}) - \boldsymbol{x}'\| \leq \varepsilon$ then $\boldsymbol{x}_i$ is safe for all $0 \leq i \leq H$. By Assumption 2, $\|M(\boldsymbol{x}, \boldsymbol{u}) - \boldsymbol{x}'\| \leq \varepsilon$ with probability at least $1 - \delta$. Then at each time step, with probability at most $\delta$ the assumption of Lemma 1 is violated. Therefore after $i$ time steps, the probability that Lemma 1 can be applied is at least $(1 - \delta)^i$, so $\boldsymbol{x}_i$ is safe with probability at least $(1 - \delta)^i$. $\qquad\square$

In order to establish a regret bound, we will analyze Algorithm 1 as a functional mirror descent in the policy space. In this view, we assume the existence of a class $\mathcal{G}$ of safe policies, a class $\mathcal{F} \supseteq \mathcal{G}$ of neural policies, and a class $\mathcal{H}$ of mixed, neurosymbolic policies.

We define a safety indicator $Z$ which is one whenever $\text{WPSHIELD}(M, \boldsymbol{x}, \pi(\boldsymbol{x})) = \pi(\boldsymbol{x})$ and zero otherwise. We will need a number of additional assumptions:

1. $\mathcal{H}$ is a vector space equipped with an inner product $\langle \cdot, \cdot \rangle$ and induced norm $\|\pi\| = \sqrt{\langle \pi, \pi \rangle}$;
2. The long-term reward $R$ is $L_R$-Lipschitz;
3. $F$ is a convex function on $\mathcal{H}$, and $\nabla F$ is $L_F$-Lipschitz continuous on $\mathcal{H}$;
4. $\mathcal{H}$ is bounded (i.e., $\sup\{\|\pi - \pi'\| \mid \pi, \pi' \in \mathcal{H}\} < \infty$);
5. $\mathbb{E}[1 - Z] \leq \zeta$, i.e., the probability that the shield modifies the action is bounded above by $\zeta$;
6. the bias introduced in the sampling process is bounded by $\beta$, i.e., $\|\mathbb{E}[\widehat{\nabla}_{\mathcal{F}} \mid \pi] - \nabla_{\mathcal{F}} R(\pi)\| \leq \beta$, where $\widehat{\nabla}_{\mathcal{F}}$ is the estimated gradient;
7. for $\boldsymbol{x} \in \mathcal{S}$, $\boldsymbol{u} \in \mathcal{A}$, and policy $\pi \in \mathcal{H}$, if $\pi(\boldsymbol{u} \mid \boldsymbol{x}) > 0$ then $\pi(\boldsymbol{u} \mid \boldsymbol{x}) > \delta$ for some fixed $\delta > 0$;
8. the KL-divergence between the true environment dynamics and the model dynamics are is bounded by $\epsilon_m$; and
9. the TV-divergence between the policy used to gather data and the policy being trained is bounded by $\epsilon_\pi$.

For the following regret bound, we will need three useful lemmas from prior work. These lemmas are reproduced below for completeness.

**Lemma 2.** *(Janner et al. (2019), Lemma B.3) Let the expected KL-divergence between two transition distributions be bounded by $\max_t \mathbb{E}_{\boldsymbol{x} \sim p_1^t(\boldsymbol{x})} D_{KL}(p_1(\boldsymbol{x}'\boldsymbol{u} \mid \boldsymbol{x})\|p_2(\boldsymbol{x}', \boldsymbol{u} \mid \boldsymbol{x})) \leq \epsilon_m$ and $\max_{\boldsymbol{x}} D_{TV}(\pi_1(\boldsymbol{u} \mid \boldsymbol{x})\|\pi_2(\boldsymbol{u} \mid \boldsymbol{x})) < \epsilon_\pi$. Then the difference in returns under dynamics $p_1$ with policy $\pi_1$ and $p_2$ with policy $\pi_2$ is bounded by*

$$|R_{p_1}(\pi_1) - R_{p_2}(\pi_2)| \leq \frac{2R\gamma(\epsilon_\pi + \epsilon_m)}{(1 - \gamma)^2} + \frac{2R\epsilon_\pi}{1 - \gamma} = O(\epsilon_\pi + \epsilon_m).$$

**Lemma 3.** *(Anderson et al. (2020), Appendix B) Let $D$ be the diameter of $\mathcal{H}$, i.e., $D = \sup\{\|\pi - \pi'\| \mid \pi, \pi' \in \mathcal{H}\}$. Then the bias incurred by approximating $\nabla_{\mathcal{H}} R(\pi)$ with $\nabla_{\mathcal{F}} R(\pi)$ is bounded by*

$$\left\|\mathbb{E}\left[\hat{\nabla}_{\mathcal{F}} \mid \pi\right] - \nabla_{\mathcal{H}} R(\pi)\right\| = O(\beta + L_R\zeta)$$

**Lemma 4.** *(Verma et al. (2019), Theorem 4.1) Let $\pi_1, \ldots, \pi_T$ be a sequence of safe policies returned by Algorithm 1 (i.e., $\pi_i$ is the result of calling WPSHIELD on the trained policy) and let $\pi^*$ be the optimal safe policy. Letting $\beta$ and $\sigma^2$ be bounds on the bias and variance of the gradient*

*estimation and let $\epsilon$ be a bound on the error incurred due to imprecision in* WPSHIELD. *Then letting $\eta = \sqrt{\frac{1}{\sigma^2}\left(\frac{1}{T} + \epsilon\right)}$, we have the expected regret over $T$ iterations:*

$$R(\pi^*) - \mathbb{E}\left[\frac{1}{T}\sum_{i=1}^{T} R(\pi_i)\right] = O\left(\sigma\sqrt{\frac{1}{T} + \epsilon} + \beta\right).$$

Now using Lemma 2, we will bound the gradient bias incurred by using model rollouts rather than true-environment rollouts.

**Lemma 5.** *For a given policy $\pi$, the bias in the gradient estimate incurred by using the environment model rather than the true environment is bounded by*

$$\left|\hat{\nabla}_{\mathcal{F}}R(\pi) - \nabla_{\mathcal{H}}R(\pi)\right| = O(\epsilon_m + \epsilon_\pi).$$

*Proof.* Recall from the policy gradient theorem (Sutton et al., 1999) that

$$\nabla_{\mathcal{F}}R(\pi) = \mathbb{E}_{\boldsymbol{x}\sim\rho_\pi, \boldsymbol{u}\sim\pi}\left[\nabla_{\mathcal{F}}\log\pi(\boldsymbol{u} \mid \boldsymbol{x})Q^\pi(\boldsymbol{x}, \boldsymbol{u})\right]$$

where $\rho_\pi$ is the state distribution induced by $\pi$ and $Q^\pi$ is the long-term expected reward starting from state $\boldsymbol{x}$ under action $\boldsymbol{u}$. By Lemma 2, we have $|Q^\pi(\boldsymbol{x}, \boldsymbol{u}) - \hat{Q}^\pi(\boldsymbol{x}, \boldsymbol{u})| \leq O(\epsilon_m + \epsilon_\pi)$ where $\hat{Q}^\pi$ is the expected return under the learned environment model. Then because $\log\pi(\boldsymbol{u} \mid \boldsymbol{x})$ is the same regardless of whether we use the environment model or the true environment, we have $\nabla_{\mathcal{F}}\log\pi(\boldsymbol{u} \mid \boldsymbol{x}) = \hat{\nabla}_{\mathcal{F}}\log\pi(\boldsymbol{u} \mid \boldsymbol{x})$ and

$$\begin{aligned}
\left|\hat{\nabla}_{\mathcal{F}}R(\pi) - \nabla_{\mathcal{F}}R(\pi)\right| &= \left|\mathbb{E}\left[\hat{\nabla}_{\mathcal{F}}\log\pi(\boldsymbol{u} \mid \boldsymbol{x})\hat{Q}^\pi(\boldsymbol{x}, \boldsymbol{u})\right] - \mathbb{E}\left[\nabla_{\mathcal{F}}\log\pi(\boldsymbol{u} \mid \boldsymbol{x})Q^\pi(\boldsymbol{x}, \boldsymbol{u})\right]\right| \\
&= \left|\mathbb{E}\left[\nabla_{\mathcal{F}}\log\pi(\boldsymbol{u} \mid \boldsymbol{x})\hat{Q}^\pi(\boldsymbol{x}, \boldsymbol{u})\right] - \mathbb{E}\left[\nabla_{\mathcal{F}}\log\pi(\boldsymbol{u} \mid \boldsymbol{x})Q^\pi(\boldsymbol{x}, \boldsymbol{u})\right]\right| \\
&= \left|\mathbb{E}\left[\nabla_{\mathcal{F}}\log\pi(\boldsymbol{u} \mid \boldsymbol{x})\hat{Q}^\pi(\boldsymbol{x}, \boldsymbol{u}) - \nabla_{\mathcal{F}}\log\pi(\boldsymbol{u} \mid \boldsymbol{x})Q^\pi(\boldsymbol{x}, \boldsymbol{u})\right]\right| \\
&= \left|\mathbb{E}\left[\nabla_{\mathcal{F}}\log\pi(\boldsymbol{u} \mid \boldsymbol{x})\left(\hat{Q}^\pi(\boldsymbol{x}, \boldsymbol{u}) - Q^\pi(\boldsymbol{x}, \boldsymbol{u})\right)\right]\right|
\end{aligned}$$

Now because we assume $\pi(\boldsymbol{u} \mid \boldsymbol{x}) > \delta$ whenever $\pi(\boldsymbol{u} \mid \boldsymbol{x}) > 0$, the gradient of the log is bounded above by a constant. Therefore,

$$\left|\hat{\nabla}_{\mathcal{F}}R(\pi) - \nabla_{\mathcal{H}}R(\pi)\right| = O(\epsilon_m + \epsilon_\pi).$$

$\square$

**Theorem 2.** (SPICE *converges to an optimal safe policy.*) *Let $\pi_S^{(i)}$ for $1 \leq i \leq T$ be a sequence of safe policies learned by* SPICE *(i.e., $\pi_S^{(i)} = \lambda\boldsymbol{x}.\text{WPSHIELD}(M, \boldsymbol{x}, \pi(\boldsymbol{x})))$ and let $\pi_S^*$ be the optimal safe policy. Let $\beta$ and $\sigma^2$ be the bias and variance in the gradient estimate which is incurred due to sampling. Then setting the learning rate $\eta = \sqrt{\frac{1}{\sigma^2}\left(\frac{1}{T} + \epsilon\right)}$, we have the expected regret bound:*

$$R(\pi_S^*) - \mathbb{E}\left[\frac{1}{T}\sum_{i=1}^{T} R\left(\pi_S^{(i)}\right)\right] = O\left(\sigma\sqrt{\frac{1}{T} + \epsilon} + \beta + L_R\zeta + \epsilon_m + \epsilon_\pi\right)$$

*Proof.* The total bias in gradient estimates is bounded by the sum of (i) the bias incurred by sampling, (ii) the bias incurred by shield interference, and (iii) the bias incurred by using an environment model rather than the true environment. Part (i) is bounded by assumption, part (ii) is bounded by Lemma 3, and part (iii) is bounded by Lemma 5. Combining these results, we find that the total bias in the gradient estimate is $O(\beta + L_R\zeta + \epsilon_m + \epsilon_\pi)$. Plugging this bound into Lemma 4, we reach the desired result. $\square$

## B    Overlapping Polyhedra

In Section 4.3, we claimed that in many environments, the safe region can be represented by *overlapping* polyhedra. In this section, we formalize the notion of "overlapping" in our context and explain why many practical environments satisfy this property.

We say two polyhedra "overlap" if their intersection has positive volume. That is, polyhedra $p_1$ and $p_2$ overlap if $\mu(p_1 \cap p_2) > 0$ where $\mu$ is the Lebesgue measure.

Often in practical continuous control environments, either this property is satisfied, or it is impossible to verify any safe trajectories at all. This because in continuous control, the system trajectory is a path in the state space, and this path has to move between the different polyhedra defining the safe space. To see how this necessitates our overlapping property, let's take a look at a few possibilities for how the path can pass from one polyhedron $p_1$ to a second polyhedron $p_2$. For simplicity, we'll assume the polyhedra are closed, but this argument can be extended straightforwardly to open or partially open polyhedra.

- If the two polyhedra are disconnected, then the system is unable to transition between them because the system trajectory must define a path in the safe region of the state space. Since the two sets are disconnected, the path must pass through the unsafe states, and therefore cannot be safe.
- Suppose the dimension of the state space is $n$ and the intersection of the two polyhedra is an $n - 1$ dimensional surface (for example, if the state space is 2D then the polyhedra intersect in a line segment). In this case, we can add a new polyhedron to the set of safe polyhedra in order to provide an overlap to both $p_1$ and $p_2$. Specifically, let $X$ be the set of vertices of $p_1 \cap p_2$. Choose a point $x_1$ in the interior of $p_1$ and a point $x_2$ in the interior of $p_2$. Now define $p'$ as the convex hull of $X \cup \{x_1, x_2\}$. Note that $p' \subseteq p_1 \cup p_2$, so we can add $p'$ to the set of safe polyhedra without changing the safe state space as a whole. However, $p'$ overlaps with both $p_1$ and $p_2$, and therefore the modified environment has the overlapping property.
- Otherwise, $p_1 \cap p_2$ is a lower-dimensional surface. Then for every point $x \in p_1 \cap p_2$ and for every $\epsilon > 0$ there exists an unsafe point $x'$ such that $\|x - x'\| < \epsilon$. In order for the system to transition from $p_1$ to $p_2$, it must pass through a point which is arbitrarily close to unsafe. As a result, the system must be arbitrarily fragile — any perturbation can result in unsafe behavior. Because real-world systems are subject to noise and/or modeling error, it would be impossible to be sure the system would be safe in this case.

## C    Further Experimental Data

In this section we provide further details about our experimental setup and results.

Our experiments are taken from Anderson et al. (2020), and consist of 10 environments with continuous state and action spaces. The mountain-car and pendulum benchmarks are continuous versions of the corresponding classical control environments. The acc benchmark represents an adaptive cruise control environment. The remaining benchmarks represent various situations arising from robotics. See Anderson et al. (2020), Appendix C for a more complete description of each benchmark.

As mentioned in Section 6, our tool is built on top of MBPO (Janner et al., 2019) using SAC (Haarnoja et al., 2018a) as the underlying learning algorithm. We gather real data for 10 episodes for each model update then collect data from 70 simulated episodes before updating the environment model again. We look five time steps into the future during safety analysis. Our SAC implementation (adapted from Tandon (2018)) uses automatic entropy tuning as proposed in Haarnoja et al. (2018b). To compare with CPO we use the original implementation from Achiam et al. (2017). Each training process is cut off after 48 hours. We train each benchmark starting from nine distinct seeds.

Because the code for Bharadhwaj et al. (2021) is not available, we use a modified version of our code for comparison, which we label CSC-MBPO. Our implementation follows Algorithm 1 except that WPSHIELD is replaced by an alternative shielding framework. This framework learns a neural safety signal using conservative Q-learning and then resamples actions from the policy until a safe action is chosen, as described in Bharadhwaj et al. (2021). We chose this implementation in order to

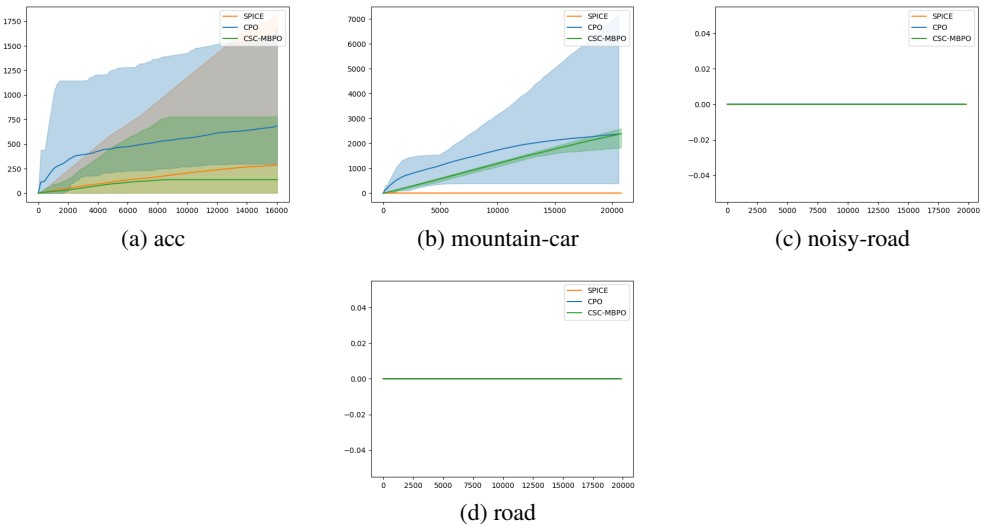

Figure 5: Cumulative safety violations over time.

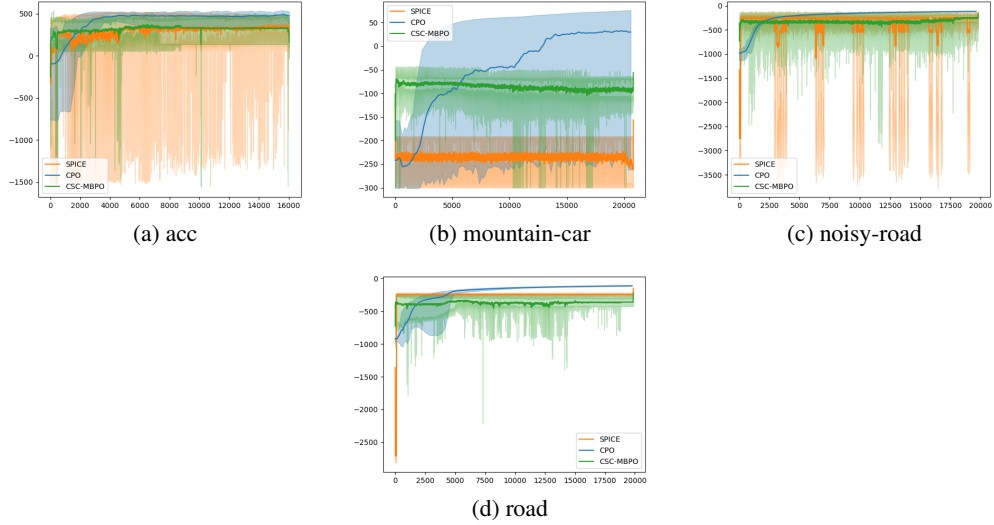

Figure 6: Training curves for SPICE and CPO.

give the fairest possible comparison between SPICE and the conservative safety critic approach, as the only differences between the two tools in our experiments is the shielding approach. The code for our tool includes our implementation of CSC-MBPO.

The safety curves for the remaining benchmarks are presented in Figure 5.

Training curves for the remaining benchmarks are presented in Figure 6.

## C.1 EXPLORING THE SAFETY HORIZON

As mentioned in Section 6, SPICE relies on choosing a good horizon over which to compute the weakest precondition. We will now explore this tradeoff in more detail. Safety curves for each benchmark under several different choices of horizon are presented in Figure 7. The performance curves for each benchmark are shown in Figure 8.

There are a few notable phenomena shown in these curves. As expected, in most cases using a safety horizon of one does not give particularly good safety. This is expected because as the safety horizon

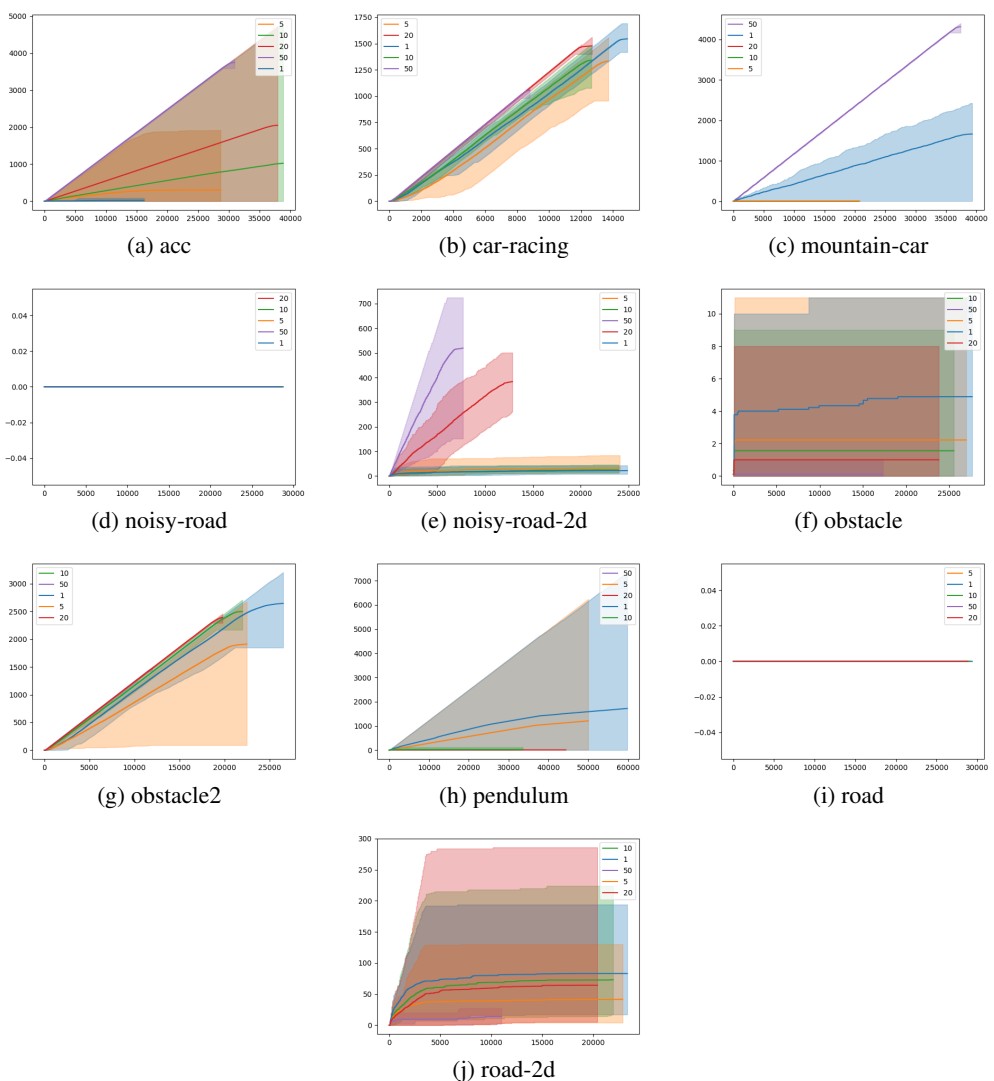

Figure 7: Safety curves for SPICE using different safety horizons

becomes very small, it is easy for the system to end up in a state where there are no safe actions. The obstacle benchmark shows this trend very clearly: as the safety horizon increases, the number of safety violations decreases.

On the other hand, several benchmarks (e.g., acc, mountain-car, and noisy-road-2d) show a more interesting dynamic: very large safety horizons also lead to an increase in safety violations. This is a little less intuitive because as we look farther into the future, we should be able to avoid more unsafe behaviors. However, in reality there is an explanation for this phenomenon. The imprecision in the environment model (both due to model learning and due to the call to APPROXIMATE) accumulates for each time step we need to look ahead. As a result, large safety horizons lead to a fairly imprecise analysis. Not only does this interfere with exploration, but it can also lead to an infeasible constraint set in the shield. (That is, $\phi$ in Algorithm 2 becomes unsatisfiable.) In this case, the projection in Algorithm 2 is ill-defined, so SPICE relies on a simplistic backup controller. This controller is not always able to guarantee safety, leading to an increase in safety violations as the horizon increases.

In practice, we find that a safety horizon of five provides a good amount of safety in most benchmarks without interfering with training. Smaller or larger values can lead to more safety violations while also reducing performance a little in some benchmarks. In general, tuning the safety horizon for each benchmark can yield better results, but for the purposes of this evaluation we have chosen to use the same horizon throughout.

## C.2 QUALITATIVE EVALUATION

Figure 9 shows trajectories sampled from each tool at various stages of training. Specifically, each 100 episodes during training, 100 trajectories were sampled. The plotted trajectories represent the worst samples from this set of 100. The environment represents a robot moving in a 2D plane which must reach the green shaded region while avoiding the red crosshatched region. (Notice that while the two regions appear to move in the figure, the are actually static. The axes in each part of the figure change in order to represent the entirety of the trajectories.) From this visualization, we can see that SPICE is able to quickly find a policy which safely reaches the goal every time. By contrast, CSC-MBPO requires much more training data to find a good policy and encounters more safety violations along the way. CPO is also slower to converge and more unsafe than SPICE.

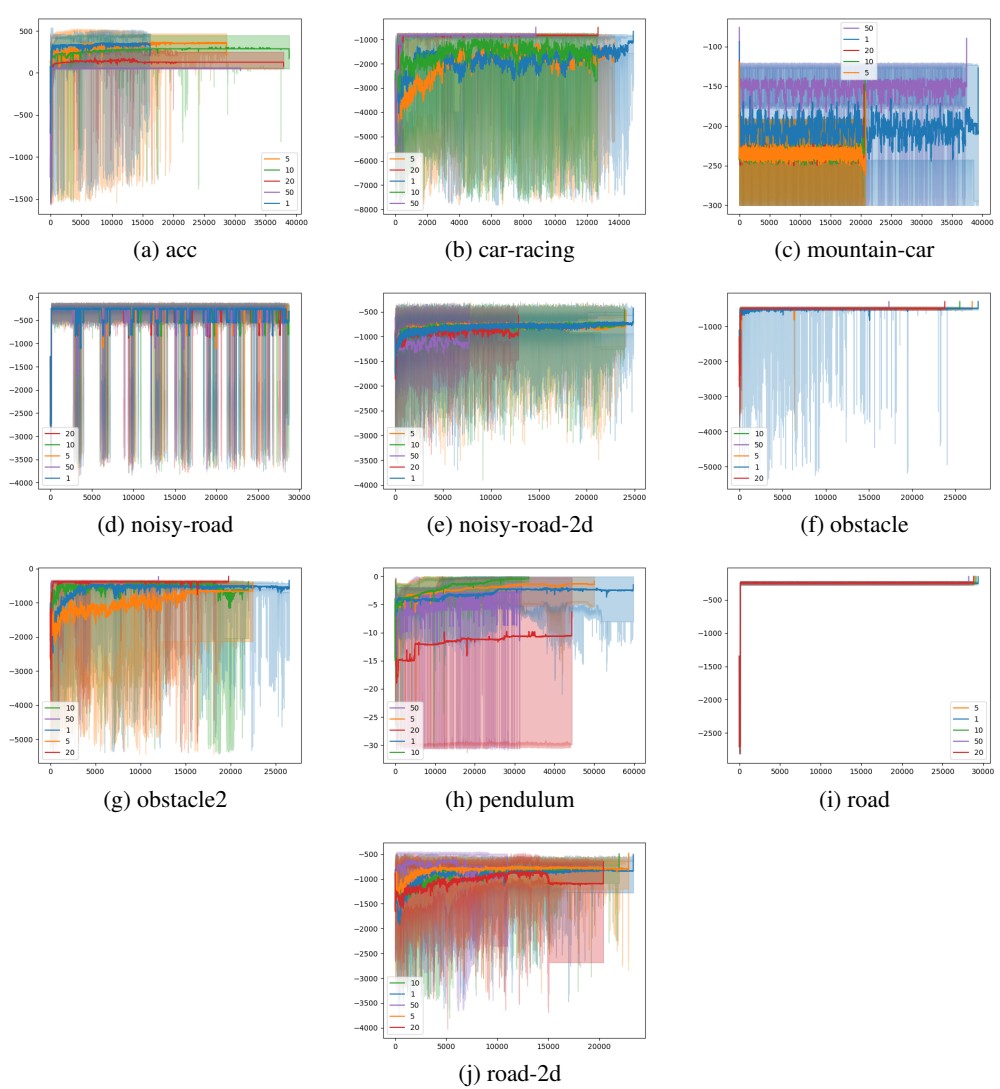

Figure 8: Training curves for SPICE using different safety horizons

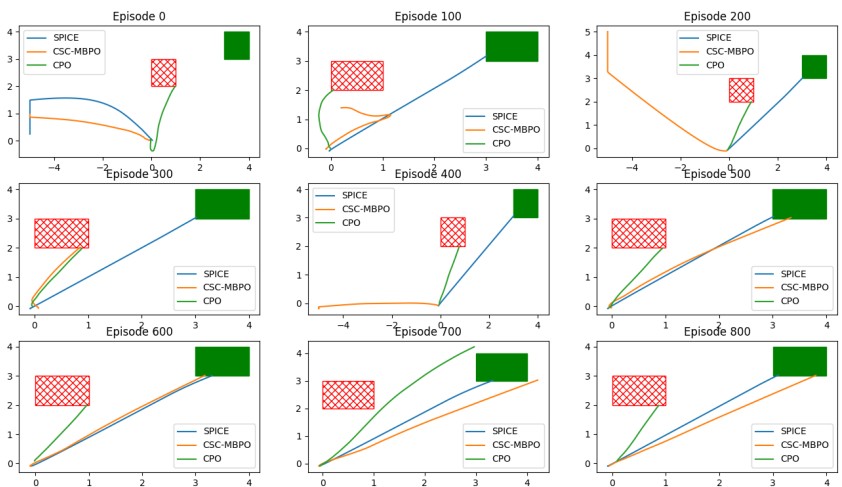

Figure 9: Trajectories at various stages of training.

