# OpenReview forum: "Guiding Safe Exploration with Weakest Preconditions"
_ICLR.cc/2023/Conference — ICLR 2023 poster_

### Official Review · Reviewer_aN3N · 2022-10-20

**Confidence:** 4
**Correctness:** 2
**Technical Novelty And Significance:** 3
**Empirical Novelty And Significance:** 2
**Recommendation:** 6

**Clarity, Quality, Novelty And Reproducibility:**

### Clarity
This paper is mostly well-written and easy to follow.

### Quality
I have several concerns regarding the quality of experiments and method itself. Especially, I am not fully convinced whether this proposed method is really useful or not.

### Novelty
As far as I know, the proposed approach using weakest precondition is new.

### Reproducibility
The source-doe is attached, and reproducibility is high.

**Strength And Weaknesses:**

### Strength
- Interesting and importance problem formulation on safe exploration.
- Unique approach for safe exploration. As far as I know, there is no existing method using weakest precondition.

### Weakness
- Though the number and quality of baselines looks sufficient to me, benchmark problem is very easy. I think the authors should have tested their method in MuJoCo or SafetyGym benchmarks. The proposed method depends on environment approximation; hence, it is particularly important to know whether it is applicable to non-linear complicated system (I think even negative results are ok, and it is significant to know the limitations).
- Low applicability to real problems. As the authors discuss the experimental results in the first paragraph on page 8, the proposed method works better than baselines when the environment can be approximated with linear equations. I think this is a huge limitation of the proposed method. I personally recommend the authors to connect this work with Linear MDP (https://arxiv.org/pdf/1907.05388.pdf), which would provide much richer theoretical results and good empirical results under the assumption of feature mapping functions.

### Questions
- How did the author choose weakest preconditions for each experiments? Is it possible to decrease the constraints violation while tuning the parameter of the weakest preconditions?

### Other comments
- The equation in Section 2 should be rewritten
    - (Section 2) $\text{Safe}(\pi)$ is not defined.
    - Is $P_{x∼S{_{\pi_i}}}(x \in S_U ) < \delta$ really what the authors want to represent? I think the probability of an agent's trajectory being safe is $(1-\delta)^N$ when the length of episode is $N$. I guess the authors should have represented in other equations.

**Summary Of The Paper:**

This paper deals with safe exploration in reinforcement learning in which an agent is required to ensure safety during training. The authors present a neuro-symbolic approach called SPICE. based on symbolic weakest preconditions. Empirically, they evaluate their approach on toy benchmarks, and show that it is able to achieve comparable performance to existing safe learning techniques while incurring fewer safety violations.

**Summary Of The Review:**

Though this paper presents a new, promising method for an interesting problems, there are several issues and concerns about the applicability and empirical evaluations. Hence, I recommend rejection for now.

---

### Official Review · Reviewer_PYp3 · 2022-10-24

**Confidence:** 3
**Correctness:** 4
**Technical Novelty And Significance:** 3
**Empirical Novelty And Significance:** 3
**Recommendation:** 8

**Clarity, Quality, Novelty And Reproducibility:**

The technique and motivation are clearly communicated and the algorithm is (to my knowledge) distinct from other shielding techniques. While the individual building blocks are well established, this seems to be a novel application well worth studying.

**Strength And Weaknesses:**

# Strengths

The technique is well motivated and a good proof of concept regarding combining model based safety analysis with model free RL. There is a clear path towards having robust theoretical guarantees (see weaknesses below) and a there is a literature of techniques on abstract interpretation that are likely to improve the representations of the preconditions.

# Weaknesses

1. Some of the assumptions in the theorems seems difficult to realize. In particular, because the learned safety shield biases the exploration, it seems difficult to assert that the final model will always achieve a given PAC bound. Perhaps I am missing something.

2. The analysis is done using a linearization of the model. This limits the class of models for which the shield can accurately forecast safety. This is acknowledged in the paper and the proposed directions for mitigation seem reasonable.

3. Missing from the related work seems to the literature on abstract interpretation that (from my outsider knowledge) offers a similar promise of learning pre-conditions for safety. Since the preconditions considered in this work are (under?) approximations for the learned model, it seems good to compare.

**Summary Of The Paper:**

This paper adapts ideas from formal program analysis, i.e. weakest preconditions, to the constrained  reinforcement learning. Here an emphasis is made for (i) working in an a-priori unknown MDP and (ii) minimizing constraint violations during deployment *and* training.

The key idea is to alternate (i) shielded exploration (ii) system identification/ model learning and (iii) computing a new avoid predicate for the shield. Again, this is done by adapting a classic idea from program analysis (called weakest preconditions) where one tries to identify the largest set such of inputs (the precondition) that guarantees the property (here not violating the constraint) is satisfied.

Technically, this by linearizing the learned model and learning a (union) of precondition polyhedra. These polyhedra are then used to project the actions proposed by the learned control policy to a safe action (according the learned model).

Empirically this seems to be very effective and future extensions are proposed (using non-linear models) that would likely yield unambigously state-of-the-art performance.

**Summary Of The Review:**

The paper contributes a novel algorithm for creating safety shields during the training of RL-agents. It provides a nice demonstration of the power of alternating model learning and state-of-the-art RL techniques for safe RL. In particular, the paper highlights how model learning in this setting need only deal with qualitative semantics (i.e., is it possible to reach state s) and can let the RL algorithm handle the quantitative optimizations.

---

### Official Review · Reviewer_HV3z · 2022-10-25

**Confidence:** 3
**Correctness:** 4
**Technical Novelty And Significance:** 3
**Empirical Novelty And Significance:** 2
**Recommendation:** 6

**Clarity, Quality, Novelty And Reproducibility:**

The paper is overall well written and presented. The contribution includes some
novel aspects pertaining to the computation of the weakest preconditions. The
significance of the overall method is adequately evaluated.

**Strength And Weaknesses:**

+ The approach reduces the number of safety violations during training when
compared with the state-of-the-art in the area.

- Somewhat straightforward and incremental: straightforward in that it relies
  on linear approximations of the environment and incremental in that the
  contribution is limited to the derivation of safety predicates within a
  previously studied framework.

**Summary Of The Paper:**

The paper proposes a method for safe reinforcement learning whereby it uses a
learnt environment to not only optimise policies but also to improve safe
exploration. Concretely, this is realised by taking a linear approximation of
the environment, which in conjunction with the safety specification gives
linear constraints (called weakest preconditions in the paper) on the actions
that are safe to be performed.

**Summary Of The Review:**

The paper builds on previous work on safe reinforcement learning  to derive a
method which although makes strong, linearity assumptions on the environment is
able to outperform the state-of-the-art in terms of the number of safety
violations during training.

---

### Official Review · Reviewer_euEh · 2022-11-03

**Confidence:** 4
**Correctness:** 4
**Technical Novelty And Significance:** 3
**Empirical Novelty And Significance:** 2
**Recommendation:** 6

**Clarity, Quality, Novelty And Reproducibility:**

The paper is generally clear. I appreciate the use of examples to demonstrate the approach.
However, the statement of theorem 2 should be improved:
* Some of the symbols that appear in the regret bound are only defined in the appendix ($L_R$, $\sigma$) or in the following text ($\zeta$).
* To be precise, $\epsilon_m$ and $\epsilon_\pi$ are upper bounds on the divergences that hold for all $T$. (This is stated in the appendix, but not in the main text.) The actual divergences are changing throughout training as the model and policy are updated.

The proposed approach is novel, to my knowledge.

Regarding reproducibility, the lack of details regarding the CSC implementation is a significant concern.

The paper is missing some relevant references to model-based safe RL papers, such as
* Safe Reinforcement Learning Using Robust MPC. M. Zanon, S. Gros
* Safe Reinforcement Learning by Imagining the Near Future. G. Thomas, Y. Luo, T. Ma


**Strength And Weaknesses:**

Strengths:
* SPICE’s use of formal methods provides strong safety guarantees if the assumptions hold. This is very desirable in safety critical applications.
* In experiments, SPICE substantially reduces the number of violations substantially compared to CPO and CSC.

Weaknesses:
* A footnote states "We use a modified version of our approach instead of comparing to Bharadhwaj et al. (2021) directly because the code for that paper is unavailable." While this is reasonable, the present paper does not provide a description of your modifications. It is difficult for the reader to compare the performance of SPICE vs. “CSC” without knowing exactly what “CSC” means.
* From Figure 3, the variance of the policy’s performance appears extremely high for both SPICE and CSC. The plots in the CSC paper look much less noisy. This raises questions about the quality of the implementation.
* SPICE converges to a suboptimal policy (worse than CPO) in many cases.
* The experiments do not compare to any other model-based safe RL algorithms.
* Linearization of the model may limit what types of problems SPICE can solve effectively.

**Summary Of The Paper:**

The paper proposes a safe RL algorithm, Symbolic Preconditions for Constrained Exploration (SPICE), that uses a model-based shielding mechanism to produce safe actions. A linearization of the model and polyhedral state constraints are used to determine which actions are safe, and then the proposed action is projected onto the space of safe actions. The paper provides a regret bound for the algorithm and demonstrates that SPICE can attain fewer safety violations during training than model-free baselines.

**Summary Of The Review:**

I like the proposed approach and its associated guarantees. My main issues are in the experiments, as detailed above. The state of the experiments makes it hard to recommend acceptance in the paper’s current form, in spite of other positive attributes of the paper.

---

### Decision · Program_Chairs · 2023-01-20

**Decision:**

Accept: poster

**Justification For Why Not Higher Score:**

There were substantial concerns from all reviewers.

**Justification For Why Not Lower Score:**

N/A

**Metareview: Summary, Strengths And Weaknesses:**

This paper provides a novel and potentially important strategy for using the technique of weakest preconditions, from the CS verification literature, to constrain exploration during reinforcement learning, and provide safety guarantees.
By making an analytical model of the preconditions of a function, it is possible to efficiently, analytically chain the conditions to compute an h-step precondition of a safety condition.  This is used as a "shield" during exploration.

The reviewers were generally positive about the ideas in the paper but had some substantial concerns.  The major concerns were:
- Weaknesses in experimental methodology:
  o the apparent variance in the training curves for SPICE and CSC seems huge (both in main results and in appendix):  this requires some exploration and explanation
  o it might have been good, to compare your replication of CSC against their published results as a sanity check
  o it would have been good to try some harder RL problems
- It would have been good to make some of the assumptions of the theorems clearer in the main body of the paper.
  o the assumption of approximate correctness of the learned model is very strong
  o there is a general concern that the linear approximation may not model the system well enough

The discussion in the appendix about the effect of changing the horizon was interesting.  Would be nice to put some discussion of that in main paper if possible. After rebuttal, authors have addressed most of the concerns, so we recommend to accept.

**Summary Of Ac-Reviewer Meeting:**

Reviewers with more expertise in the verification community found the methods to be relatively straightforward and incremental. Reviewers with more expertise in the RL community were concerned about experimental results (high variance, simple domains), but rebuttal has addressed many of them. There was agreement that the approach was interesting.